# A Review on Resource Utilization of Spent V-W-Ti Based Selective Catalytic Reduction Catalysts

**DOI:** 10.3390/ma15227984

**Published:** 2022-11-11

**Authors:** Xi Qian, Weihua Ao, Hao Ding, Xuan Wang, Sijia Sun

**Affiliations:** Beijing Key Laboratory of Materials Utilization of Nonmetallic Minerals and Solid Wastes, National Laboratory of Mineral Materials, School of Materials Science and Technology, China University of Geosciences, Xueyuan Road, Haidian District, Beijing 100083, China

**Keywords:** SCR, photocatalyst, vanadium, tungsten, titanium, resource utilization

## Abstract

To address the environmental pollution caused by nitrogen oxides, V_2_O_5_-WO_3_/TiO_2_ is widely used as a catalyst based on selective catalytic reduction (SCR) technology. However, spent SCR catalysts pose a potential hazard to the environment due to the presence of heavy metals. This problem continues to plague countries with predominantly thermal power generation, and landfills as the dominant disposal method wastes significant metal resources. Previous research into the recovery of these metal resources has received considerable attention. Here, we summarise the methods of recovery and find that research trends are beginning to move towards improving the added value of recovered products. One very promising application is photocatalysts; however, the atomic efficiency of current methods is not satisfactory. Therefore, this review first focuses on the regeneration of spent SCR catalysts and the processes used for elemental extraction to clarify what forms of V, W and Ti can be obtained from existing processes. This is followed by providing directions for the conversion of spent SCR catalysts into photocatalysts with improvements based on such processes. From a different perspective, this also provides a new resource for photocatalysts and is expected to significantly reduce the cost of photocatalyst production.

## 1. Introduction

Industrial gas emission has been a concern for decades, and nitrogen oxides (NO_X_) are one of the most important toxic industrial gases [1,2]. A selective catalytic reduction catalyst (SCR catalyst) is widely used for coal-fired boilers as a flue gas denitrification catalyst to convert NO_X_ into harmless N_2_ emission [3]. Furthermore, V_2_O_5_-WO_3_/TiO_2_-based SCR catalysts are the most common type. At high temperatures, the presence of H_2_O, SO_2_, alkali metals, heavy metals and halogens causes the SCR catalyst to be poisoned and, after deactivation, become a spent SCR catalyst [4]. Spent SCR catalysts have several characteristics: (i) increasing waste generation [5], (ii) containing large amounts of metal resources [6], and (iii) causing serious threats to soil and water bodies [7]. The severity and urgency of the problem is reflected in the increase in patents for the recycling of spent SCR catalysts in China since 2013 [8].

The treatment process must prevent toxic elements (e.g., V, As, Pb, etc.) from entering soil and water bodies to avoid secondary pollution. Conventional treatment is landfill for spent SCR catalysts as hazardous solid waste [9]. According to the policy guidelines, such wastes are required to be roasted above 1000 °C with flux to prevent the leaching of toxic elements [10]. Recent research has tended to add spent catalysts to production lines that require significant use of SCR catalyst for denitrification, thus avoiding transportation. For instance, spent SCR catalysts can be blended into the iron-ore sintering process [11], with heavy metals being recovered from the sintering dust [12]. However, the addition of spent SCR catalysts will reduce several indicators [13]. Alternatively, the spent SCR catalyst can also be mixed into cement and the leaching of As and Pb is significantly inhibited after sintering [14].

However, spent SCR catalysts have a high recovery value with components of over 80wt% consisting of V_2_O_5_, WO_3_ and TiO_2_ and should, therefore, be regenerated or recovered in order to recycle the resource [15,16,17]. Although spent SCR catalysts may contain oxides such as SiO_2_, Al_2_O_3_ and CaO that constitute the ceramics, as well as compounds of elements including As and Pb deposited from the flue gas, the extraction of these materials is not reviewed in this article. Regeneration methods can extend the life of SCR catalysts by focusing on detoxification and reactivation. The major problems to be addressed during regeneration are the deactivation and loss of active sites as well as blocking of pores [4,18,19,20]. Specific regeneration methods for different types of deactivations can achieve promising results [21]. However, current regeneration processes are unable to achieve the expected results when the catalyst is severely deactivated or has undergone several regenerations. In this case, the SCR catalyst will reach the end of its life and be recycled by recovery [22]. Conventional recovery methods focused on the extraction and purification of metals (V, W and Ti) [23]. However, recent research has increasingly turned to the added value of the product with a view to increasing the economic benefits of recycling. For instance, after separation, NH_4_VO_3_, ammonium paratungstate and anatase TiO_2_ are produced, respectively [24].

Under light conditions, photocatalysts produce photogenerated electrons and holes and, further, form active species such as hydroxyl (·OH) and oxygen (O_2_^−^) radicals. Due to the strong redox properties of the active species, photocatalysts can address pollutions caused by heavy metals, organics and other substances [25]. Anatase TiO_2_ is a classical photocatalyst with excellent photocatalytic activity. Studies have often used doping to build heterostructures to extend the wavelength range of light and to suppress the separation of photogenerated carriers [26,27]. On the other hand, to inhibit the agglomeration of nano TiO_2_ and to assist recycling, the preference is to use carrier-loaded nano TiO_2_ [28]. Spent SCR catalysts contain a large amount of anatase TiO_2_ and are cost-effective; therefore, recovery as photocatalysts not only increases the recovery benefit, but also overcomes the high cost of conventional photocatalysts. In addition, V [29,30] and W [31,32] in spent SCR catalysts have the potential to build heterostructures with TiO_2_.

This paper will first review the research on the recycling of spent SCR catalysts carried out between 2013 and 2022. Thereafter, it will continue with a discussion of the studies on the recycling of spent SCR catalysts between 2019 and 2022, including both the regeneration and recovery of spent SCR catalysts. The discussion will focus on the use of reagents with acid, base, complexing, oxidising, or reducing properties in these studies and will summarise the different effects of the different reagents. For the recovery methods, the extraction process of the elements is discussed. This is followed by an introduction to the idea of converting spent SCR catalysts into photocatalysts. Finally, the developments in the recycling of spent SCR catalysts are concluded.

## 2. Progress in Research on Spent SCR Catalysts

### 2.1. Methodology

Articles were retrieved on 24 July 2022 from Web of Science (www.webofscience.com) database. The articles were found using the following search formula: TS = (selective-catalytic-reduction OR NH3-scr or deNO(x)-catalysts OR scr) AND TS = (spent OR waste) AND TS = (recovery OR recycling OR leaching OR extraction OR management) AND TS = (titanium OR vanadium OR tungsten). Patents were found through the European Patent Office (www.epo.org) on 24 July 2022. The keywords used are selective-catalytic-reduction, NH3 scr, deNO, scr, spent, recovery, recycling, leaching, extraction, management, titanium, vanadium, tungsten. Patents and articles between 2013 and 2022 were counted using the search method described above to give a general trend of research on recycling spent SCR catalysts. Other sections review articles for 2019 to 2022.

### 2.2. Research Process

The data on articles and patents described in 2.1 are plotted in Figure 1, where Figure 1a shows the number of articles and patents issued with the year, and Figure 1b shows the proportion of patent disclosures in different countries or organisations. From 2013 onwards, the number of studies addressing the recovery and regeneration of spent SCR catalysts continues to rise and reaches a peak in 2020. The proportion of patents is higher than that of articles, reflecting the huge demand for practical applications in this field. In Figure 1b, significant interest in the disposal of waste SCR catalysts is shown, since the energy structure in China, Korea and Japan is dominated by thermal power generation. Take China as an example: the Emission standard of air pollutants for thermal power plants (GB 13223-2011) limits NO_X_ emissions to <100 mg/m^3^. With increasingly stringent restrictions on NO_X_ emissions, the use of SCR catalysts will also increase. In view of the limited effect of regeneration on spent SCR catalysts, the cycle time of regeneration has been limited to ensure that gas emissions comply with standards [22]. As a result, there is a growing interest in the development of regeneration processes and the reuse of resources through recycling.

The principles of green chemistry and green engineering by Ziemmerman et al. [33] in 2020 also emphasise the importance of this idea of recycling. The principles point to a shift in future industry from mostly linear processes to circular processes and a shift from “waste” treatment to “waste” utilisation. In this field, the main idea is to avoid landfills in favour of efficient regeneration of spent SCR catalysts and recycling those that cannot be regenerated. Current research has explored several methods for recycling spent SCR catalysts, but these methods have not been replicated in industry, so there is still much scope for research into recycling methods. Factors limiting their application include the process, waste generation, material consumption, and equipment requirements and, in the case of recycled production products, the value of the product is also important [22].

## 3. Regeneration of Spent SCR Catalysts

SCR catalysts under a high-temperature flue gas condition are subjected to deactivation due to multiple factors including pore blocking, catalytic-site poison and catalytic-component loss [4,18,19,20]. In recent years, most regeneration processes have employed ultrasonic instead of stirring to obtain better results (Table 1). Regeneration processes have been designed to address the poisoning of spent SCR catalysts and to attempt to restore the structure of spent SCR catalysts. Following this effort, some studies have supplemented the regenerated catalysts with active sites by impregnation. NH_4_VO_3_ solutions are usually used to reload V_2_O_5_ [34], and, relatively, Ce(NO_3_)_3_ solutions can be used instead to load CeO_2_ avoiding the use of V [35].

Improvement of the pore landscape of regenerated SCR catalysts can effectively increase catalytic activity without reloading the active site. Various data can be used to characterise the pore landscape, such as: mesopore volume, mesopore surface area, BET surface area, acidic sites etc. Here, the mesopore volume data obtained with thermoporometry measurements (TPM) provides a reliable indication of the effect of the immersing method [36]. In addition to regenerating the activity by improving the pore landscape, pore size and specific surface area can also be enhanced by loading the regenerated SCR catalyst powder onto the cloth surface. Shi et al. [34] regenerated the spent SCR catalyst in H_2_C_2_O_4_ and NH_4_VO_3_ solutions after sequential blowing, grinding, deionised water washing and 0.5 M sulphuric acid washing. The regenerated catalyst was then loaded onto a P84 filter cloth by impregnation and used as a catalytic filter for low-temperature flue gas denitrification achieving 91.6% of the catalytic effect of the fresh catalyst.

The regeneration process attempts to achieve better results for catalysts that are severely deactivated as well as repeatedly experiencing deactivation–regeneration [22]. Complex processes can effectively extend the life cycle of SCR catalysts. However, the complexity of the process also results in higher costs. Therefore, despite the fact that complex processes can be used for longer cycle times, the difficult regenerated spent SCR catalysts need to be further recovered as metal (V, W, and Ti) resources to reduce costs.

## 4. Extraction of Elements from Spent SCR Catalysts

Two common methods can be used to effectively extract V and W from spent SCR catalysts: the leaching method [39,40] and the roasting method [16,41]. The leaching method is a process dissolving certain components from the catalyst. Leaching reagents are typically employed to selectively separate specific components with acid [42], base [43], complexation [44], reduction [44], oxidation [45] and other properties. In most cases, TiO_2_ will be retained in the residue to be recycled [46,47]. The roasting method is a process in which reagents eutectic with spent SCR catalyst under high-temperature conditions form salts. This process generally forms soluble salts which are subsequently separated from the solid phase in a leaching step [48,49]. Table 2 summarises the studies of extracting by roasting or leaching methods for V, W and lists the final form of Ti.

### 4.1. Leaching Method

Leaching is a process of separating V_2_O_5_, WO_3_ from the catalyst carrier using a solvent to break the V-O, W-O and Ti-O bonds [52]. The separation can be achieved by selectively dissolving V_2_O_5_, WO_3_ and TiO_2_ into different solvents based on the difference in solubility. Whereas V_2_O_5_ is usually well soluble in acids and bases, WO_3_ is only soluble in bases and stable to acids, while TiO_2_ is always retained in residues. Therefore, leaching with normal acid (e.g., H_2_SO_4_ [52]) allows a highly selective separation of V from the catalyst, and, conversely, extracts both V and W.

In the case of V extraction, V(III) is not easily soluble and can be oxidised by adding an oxidant or V(V) under an acidic condition. H_2_O_2_ is commonly used as an oxidant to convert low-valent V to V(V) under alkaline conditions, contributing to the leaching efficiency in both leaching [45] and precipitation [54] processes. In contrast, the reaction between V(III) and V(V) will take place at 5wt% H_2_SO_4_ at 95 °C and will eventually be converted to VOSO_4_ in the presence of the reducing agent Na_2_SO_3_ (leaching efficiency is nearly 100%) [57]. Theoretical calculations show [52] that (i) with alkaline reagents, OH^-^ reacts directly with V and W atoms; and (ii) with acidic reagents, H^+^ reacts with O atoms. Organic acids such as H_2_C_2_O_4_ exhibit complexation reaction activity as well, e.g., V can be separated out as VOC_2_O_4_ using 1.0 mol/L H_2_C_2_O_4_ at 90 °C in a reaction of 3 h at a liquid-to-solid ratio of 20 mL g^−1^ (leaching efficiency of 84.22%) [44]. Among the organic acids, H_2_C_2_O_4_ is more effective than citric and tartaric acids for the extraction of V [58].

The extraction of W was also facilitated by the addition of the oxidising agent H_2_O_2_ but the leaching efficiency remained at a low level [54]. Additionally, at high V content, the leaching efficiency of W is limited due to the strong V-W-Ti interaction [50]. WO_3_ has good stability in acid and requires alkaline reagents for dissolution. Leaching of W generally uses NaOH as the leaching agent and requires heating. For instance, heating a 1.5 mol L^−1^ NaOH solution to 100 °C at atmospheric pressure for a 4 h leaching (liquid-to-solid ratio = 15) resulted in leaching efficiencies ranging from 38.0% to 57.3% for W [50]. In contrast, applying a pressure condition to heat a 4.75 mol L^−1^ NaOH solution to 190 °C for 44.5 min leaching reaction (liquid-to-solid ratio = 10) could achieve 98.63% leaching efficiency [51]. When the temperature was increased to 300 °C, 96% W (liquid-to-solid ratio = 10) was leached after 2 h of immersing using a 2 mol L^−1^ NaOH solution [39]. Further studies have found that mixing NaOH with spent SCR catalysts using a ball-mill premix promotes leaching at lower temperatures [59]. After premixing NaOH to a catalyst at a mass ratio of 0.9, leaching with the addition of water for 20 min at 25 °C (liquid-to-solid ratio = 15) could achieve the same result as a 4 h reaction at 100 °C under the same conditions (see Figure 2 for SEM images). The final leaching efficiencies for V and W were 67.7% and 56.3%, respectively.

Ammonium salt can form water-soluble NH_4_VO_3_ and (NH_4_)_2_WO_4_ with V and W, respectively. Cao et al. [45] leached V and W using a solution of 2 mol L^−1^ (NH_4_)_2_CO_3_ and 1.5 mol L^−1^ H_2_O_2_. After 0.5 h leaching at 70 °C, the leaching efficiencies of V and W reached 98% and 99% (liquid-to-solid ratio = 25). The addition of H_2_O_2_ has a promoting effect on V but is not sensitive to the concentration. The use of alkaline leaching agents requires the further separation of V and W. For example, diethylhydroxydodecanoneoxime (LIX 63) can selectively separate V from W [45]. Similar extractants are trioctylphosphine oxide (TOPO), triisobutylphosphine sulphide (TIBPS), trioctylmethylammonium chloride (Aliquat 336) and di-2-ethylhexylphosphoric acid (HDEHP) [60]. Aliquat 336 can simultaneously extract V and W [24]. After separation, W can be converted to ammonium paratungstate, while V can be converted to NH_4_VO_3_.

### 4.2. Roasting Method

Roasting is a method of converting oxides of V and W into soluble salts with the assistant of alkaline reagents by high-temperature solid-state reactions. The main reagents used are NaOH, Na_2_CO_3_ and NaCl. NaOH and Na_2_CO_3_ are the major alkaline reagents, which can react with W more effectively than NaCl [53]. The Na_2_CO_3_ will decompose above the melting temperature, accompanied by the occurrence of alkali fusion. When the CaO content is low and the Na_2_O content is high, V and W tend to form soluble NaVO_3_ and Na_2_WO_4_, thereby avoiding the formation of insoluble CaV_2_O_6_ and CaWO_4_ during the process [48]. Due to the different proportions of Na_2_O and metal oxides, the catalyst will exhibit different states after the reaction (Figure 3). Studies have shown that mixing NaCl with Na_2_CO_3_ [49] (NaCl:Na_2_CO_3_ = 8.8:16) and NaOH [53] (NaCl:NaOH = 3:2) can effectively promote leaching efficiency. NaCl will produce Cl_2_ in the process of high-temperature calcination, acting as catalyst and oxidant, thus reducing the reaction temperature (from 1000 °C to 750 °C) [53].

After roasting with an alkaline reagent, the obtained solid can be selectively extracted using the leaching method. In this condition, water is able to dissolve the sodium salts formed during roasting, including NaVO_3_, Na_2_WO_4_, etc. [49]. Thus, V and W are transferred to water. Since Si affects the leaching efficiency of V and W, Si needs to be removed first in the case of high Si content. At room temperature, 85% of Si can be removed in the form of silicate precipitation by reducing the pH to 9.5 by HCl, avoiding the loss of W and V. Water leaching can reach more than 99% leaching of V and W, and is less affected by leaching conditions [55]. Afterwards, V and W can be precipitated from water with Ca^2+^ ions [54]. Alternatively, highly selective extraction can be achieved with the help of the selective complexation of V and W by organic ammonium salts (e.g., [R_3_NCH_3_]^+^Cl^−^ [24]). Following a stripping process using a solution of NaOH and NaCl, the stripping step has a leaching efficiency of around 70% for V and W.

Leaching processes are limited for the efficient leaching of W due to the stability of the W-O bond. In contrast, roasting methods are highly efficient in breaking such stable bonds at high temperatures, thus reducing the requirement for a leaching reagent. However, roasting reagents require NaOH or Na_2_CO_3_ for alkali fusion at high temperatures, raising the equipment requirements and, therefore, making it difficult to promote. To solve this problem, the leaching process can be adjusted to reduce the reliance on alkaline reagents during roasting. In the case of leaching with HCl instead, for example, CaO can be added to the spent SCR catalyst instead of the alkaline reagent [17]. Such acid leaching takes place by the selective dissolution of the calcium salts (e.g., CaV_2_O_6_ and CaWO_4_) formed from roasting step. The use of a high concentration (4 mol L^−1^) of HCl promotes the dissolution of V and inhibits the dissolution of W, thus achieving the separation of V and W. The undissolved W will be deposited on the surface of CaWO_4_ in the form of H_2_WO_4_. The H_2_WO_4_ precipitation should be dissolved in a low-concentration (1 mol L^−1^) NaOH solution to inhibit the dissolution of Fe, Al, and V. Subsequently CaCl_2_ is added to once again precipitate W from the solution as CaWO_4_ (CaWO_4_ content was 96.1%). Using H_2_C_2_O_4_ (0.5 mol L^−1^) instead of HCl (1 mol L^−1^) was found to increase the dissolution efficiency of W as well as reduce the dissolution of V [56]. In contrast to HCl, the use of H_2_C_2_O_4_ does not produce H_2_WO_4,_ but is further converted to soluble H_2_[WO_3_(C_2_O_4_)H_2_O]. This dissolution process is accompanied by a precipitation of CaC_2_O_4_. A secondary roasting of the leach residue converts this precipitate into CaO. Further H_2_C_2_O_4_ leaching after the secondary roasting leads to an increase in the leaching of W to 87% and directly obtaining a H_2_WO_4_ product.

### 4.3. Separation of Ti

The content of Ti is the highest in spent SCR catalysts and Ti will be retained in the residue after the extraction of V and W. With similar leaching efficiencies of V and W, differences in the form of Ti are presented due to the different processes. During roasting above 500 °C, anatase TiO_2_ tends to change to rutile TiO_2_ [7] and a portion of TiO_2_ is converted to titanate [54], yet the leaching process is relatively gentle and does not affect the structure of TiO_2_ [45]. The form of Ti varies depending on the recovery method (Figure 4): (i) roasting with Na_2_CO_3_ and NaCl at 750 °C gives a mixture of rutile TiO_2_, anatase TiO_2_ and sodium titanate [49]; (ii) roasting with K_2_CO_3_ gives potassium titanate [54]; (iii) electrolysis of Ti_2_CO (obtaining by a carbothermal reduction) gives Ti metal [6]; and (iv) adding hot concentrated HCl to the roasting residue gives a TiOCl_2_ solution [55], etc.

## 5. Conversion of Spent SCR Catalysts into Photocatalysts

Recent research has explored new recycling methods to convert spent SCR catalysts into products rather than raw materials, such as a pigment [61] and ceramic opacifier [62]. The idea of recycling spent SCR catalysts as photocatalysts for pollution treatment is in-line with the concept of “treating waste with waste” [63].

### 5.1. Catalyst Carriers

Photocatalytic degradation is a promising technology to address environmental pollution with economic and environmental efficiency [64]. However, as most photocatalysts are nanoparticles, they are difficult to recycle and, therefore, present the potential risk of secondary pollution [65]. On the other hand, the tendency of nanoparticles to agglomerate can also limit their catalytic activity [66]. To solve these problems, photocatalysts can be loaded onto carriers, such as fly ash [67] and activated carbon [68]. The carriers not only provide dispersion and easier recycling, but specific carriers can also improve the catalytic activity of the catalyst by forming a specific heterostructure [69].

Spent SCR catalysts contain a variety of oxides with catalytic activity, which can be used as carriers to enhance catalytic activity. However, it is important to ensure that components such as V_2_O_5_ and As_2_O_5_ do not enter the environment during utilization and cause secondary contamination. Jin et al. [70] investigated spent SCR catalysts by grinding into powder and adding Al_2_O_3_, diatomite and agglomerant to calcine at 1000 °C to obtain ceramics. This was followed by impregnation and sintering in an 8wt% Ni(NO_3_)_2_ solution to obtain NiO loading (Figure 5). The prepared NiO-based catalysts were used for the reforming of formaldehyde and water vapour for hydrogen production with a selectivity of 100% for H_2_, 31.9% for CO and 53.2% for CO_2_ at 500 °C, and a conversion of formaldehyde above 93.0%. The analysis of the XPS data and the mechanism of the reforming reaction revealed that the presence of oxides on the surface had a positive effect on the performance of the catalysis. Based on the work of Jin et al., it can be demonstrated that spent SCR catalysts have potential as catalyst carriers. Particularly, TiO_2_ shows typical photocatalytic activity and research on recycling spent SCR catalysts as photocatalyst carriers for this feature awaits further exploration.

### 5.2. Photocatalysts

V_2_O_5_, WO_3_ and TiO_2_ from spent SCR catalysts can be used for the preparation of photocatalysts. V_2_O_5_ and WO_3_ can form crystals with cations such as Bi^3+^ in the form of VO_4_^3−^ and WO_6_^6−^, respectively [71], while TiO_2_ exists as anatase TiO_2_ mostly [72]. Zhang et al. [57] used a solution of H_2_SO_4_ and Na_2_SO_3_ to leach out V from spent SCR catalyst, followed by dissolving the Ti-containing residue in HF solution to obtain WO_3_-TiO_2_ photocatalyst after a hydrothermal reaction. In another case, V and W were leached out using a NaOH solution, followed by the addition of Bi(NO_3_)_3_ for a hydrothermal reaction to obtain BiVO_4_/Bi_2_WO_6_ photocatalysts [71]. Wang et al. [73] also used NaOH to retain Ti in the leach residue for separation. Subsequently, Na_3_PO_4_ and Mg(NO_3_)_2_ were used to precipitate Ca^2+^, SiO_4_^4−^, and PO_4_^3−^, respectively. This was followed by the precipitation of Al^3+^ and Mg^2+^ using HNO_3_ and NaOH to adjust the pH. The solution retains VO_3_^−^ and WO_4_^2−^, but additional NH_4_VO_3_ is required due to the low V. The addition of Zn(NO_3_)_2_ eventually leads to a visible light responsive Zn_3_(VO_4_)_2_/ZnWO_4_ photocatalyst by hydrothermal reaction.

Qian et al. [74] used NaOH to leach the V element from the catalyst, followed by a hydrothermal reaction using 96% H_2_SO_4_ to convert the TiO_2_ to TiOSO_4_ after roasting at 200 °C (Figure 6). V is removed after reaction with NaOH, while W remains in the residue. This method overcomes the difficulty of separating Ti and W and makes effective use of Ti. The prepared photocatalyst exhibits similar catalytic activity to P25 at a lower Ti content and demonstrates the feasibility of dissolving and converting Ti to nano TiO_2_. Furthermore, loading TiO_2_ onto the fly-ash surface also overcomes the shortcomings of nano TiO_2_, which is difficult to recover and prone to agglomeration.

Current research has focused on the separation of TiO_2_ from V_2_O_5_ and WO_3_ with some progress being achieved. Figure 7 illustrates the transformation relationships of substances in these works. Since the goal is no longer to separate and purify V, W and Ti, the process can be simplified based on current separation processes. New extraction processes can also be developed, such as the dissolution of TiO_2_ with sulphuric acid under hydrothermal conditions [75]. Mechanochemical methods also have potential applications in the preparation of TiO_2_ [76]. Ball milling as a pre-treatment has proven to be effective in the recycling of spent SCR catalysts [59]. On the other hand, both BiVO_4_ [77] and Bi_2_WO_6_ [78] can form corresponding heterostructures with TiO_2_. Moreover, V and W can be extracted directly into solution as NH_4_VO_3_ and (NH_4_)_2_WO_4_ [45], which would assist in the further synthesis of bismuth salt.

## 6. Conclusions

To recycle spent SCR catalysts as a metal resource, it is vitally important for future research to explore environmentally friendly and economically feasible recycling methods. Regeneration processes are currently employed in industry to extend the life of SCR catalysts, but catalysts that have undergone three to four regenerations are eventually abandoned as they are difficult to regenerate. Landfill is the main disposal method for spent SCR catalysts at present; although wasteful of resources, it is much easier to process. Consequently, even though the recycling process is more in-line with the principles of green chemistry, it is difficult to promote because of its complexity, high cost, low economic efficiency, and the risk of secondary pollution. Nevertheless, further exploration of the recycling process is necessary to address these shortcomings to achieve broader application.

This review of research progress in the last 3 years shows that an increasing number of studies are focusing on the added value of the product obtained after extraction. The common elemental extraction processes include leaching and roasting methods. The leaching method can achieve the extraction of elemental V at lower temperatures, and efficiency can be improved by combining with a reducing agent (e.g., Na_2_SO_3_) under acidic conditions and oxidizing agent (e.g., H_2_O_2_) under alkaline conditions. The roasting method can overcome the difficulty of extracting W by the leaching method as it converts insoluble oxides into soluble salts. Improvements to the roasting method include (i) adaptation of the roasting condition with reference to the phase diagram, (ii) addition of a Cl^−^ containing catalyst (e.g., NaCl) to reduce the temperature and time for the reaction, and (iii) application of acid leaching rather than water leaching to avoid the use of an alkaline reagent in the roasting process. With the above method, V_2_O_5_ can be recovered as CaV_2_O_6_, NH_4_VO_3_, BiVO_4_, and Zn_3_(VO_4_)_2_, while WO_3_ is recovered as CaWO_4_, ammonium tungstate, Bi_2_WO_6_, and ZnWO_4_. Additionally, both BiVO_4_/Bi_2_WO_6_, and Zn_3_(VO_4_)_2_/ZnWO_4_ and TiO_2_ can be utilized as photocatalysts for environmental management, respectively. Further research should explore photocatalyst conversion processes with higher atomic efficiency. The selective conversion of V, W and Ti into their salt or oxide can be applied in the treatment of spent SCR catalysts, and attempts can be made to combine these substances into a heterogeneous photocatalyst. In addition, new treatment technologies such as mechanochemical processes to reduce contamination during the recycling of spent SCR catalysts need to be introduced. The process should avoid high-temperature and pressure-reaction conditions and reduce the production of wastewater, waste gases and residues.

## Figures and Tables

**Figure 1 materials-15-07984-f001:**
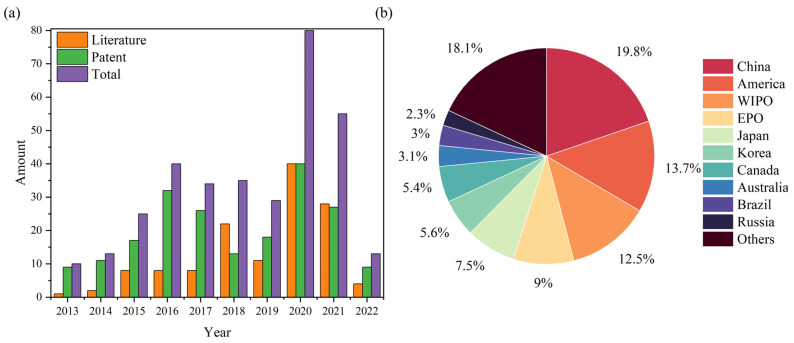
(**a**) Annual number of publications of papers and patents. (**b**) Patent issuance by countries and organizations between 2013 and 2022.

**Figure 2 materials-15-07984-f002:**
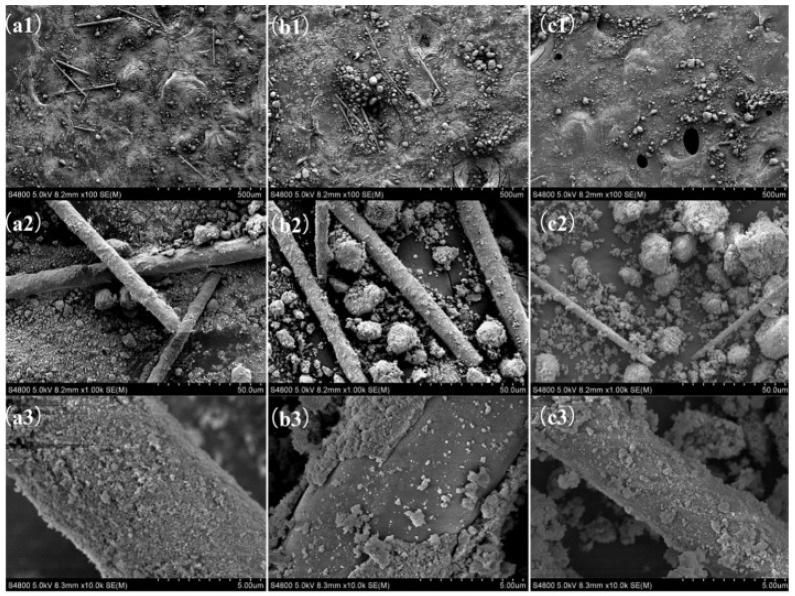
SEM images of (**a**) the original catalyst, (**b**) the catalyst treated by the premixing method and (**c**) the catalyst treated by the common method (from 1 to 3 at 100×, 1000× and 10,000×, respectively). Reproduced with permission [59]. Copyright 2018, Elsevier.

**Figure 3 materials-15-07984-f003:**
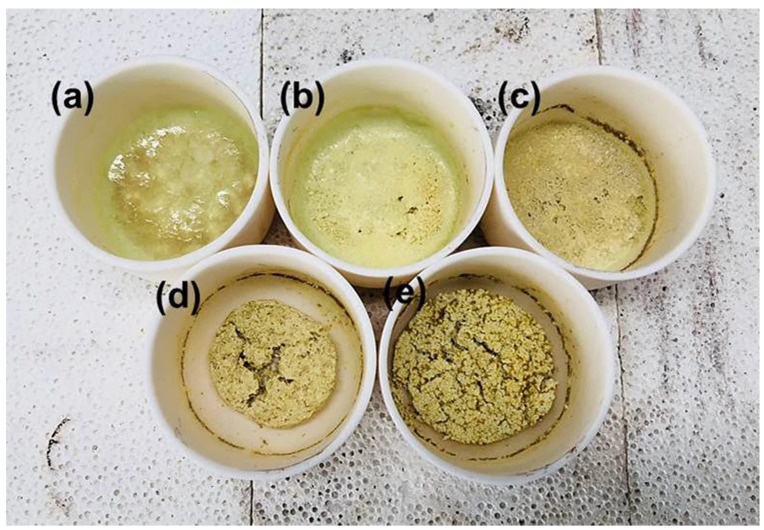
Actual appearance of samples after alkali fusion reactions for 20 min at 950 °C in various molar ratios (MO_X_/Na_2_O+MO_X_, M = Ti, Si, W): (**a**) 0.3, (**b**) 0.4, (**c**) 0.5, (**d**) 0.6, (**e**) 0.7. Reproduced with permission [48]. Copyright 2019, Springer Nature.

**Figure 4 materials-15-07984-f004:**
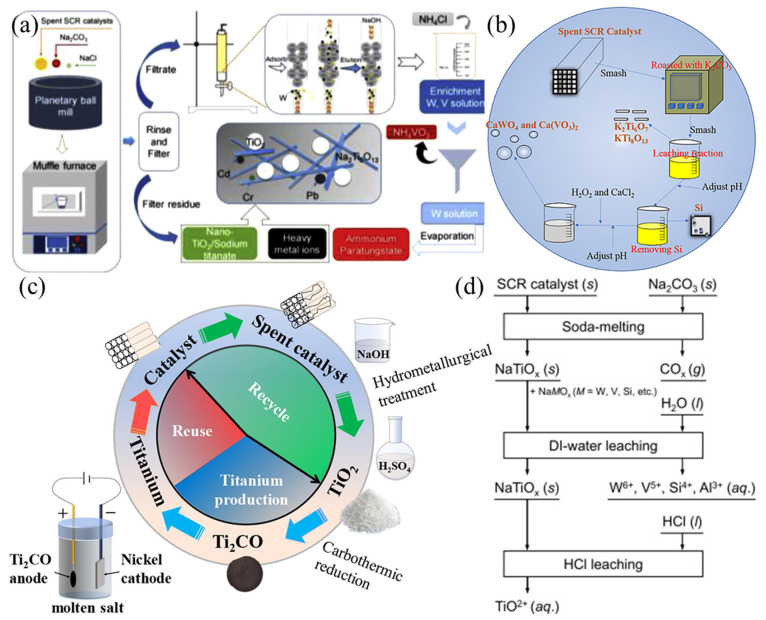
The different recovery products of Ti: (**a**) mixture of rutile TiO_2_, anatase TiO_2_ and sodium titanate. Reproduced with permission [49]. Copyright 2020, Elsevier. (**b**) Potassium titanate. Reproduced with permission [54]. Copyright 2020, Springer Nature. (**c**) Ti metal [6]. Reproduced with permission [6]. Copyright 2021, Elsevier. (**d**) TiOCl_2_ solution. Reproduced with permission [55]. Copyright 2019, Elsevier.

**Figure 5 materials-15-07984-f005:**
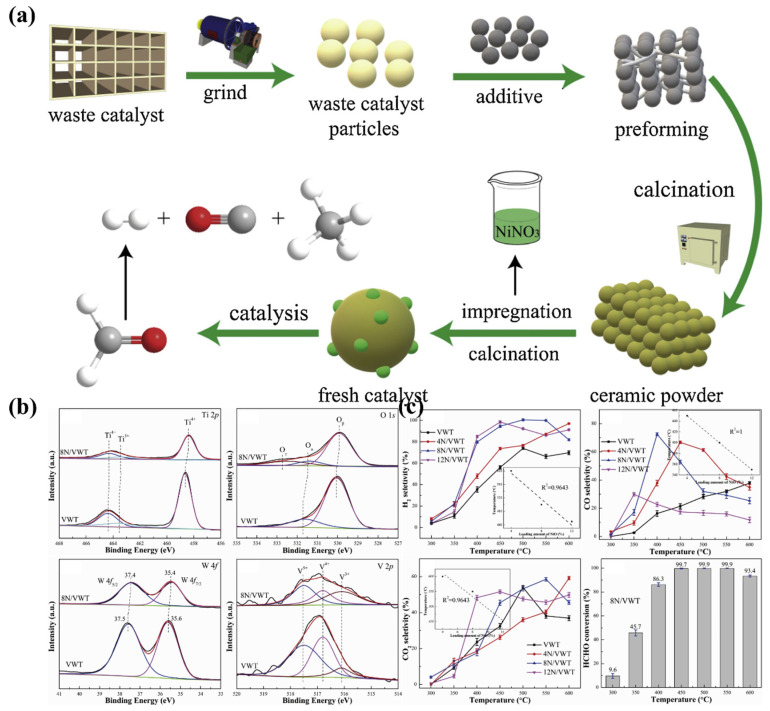
(**a**) Process of converting spent SCR catalysts into NiO-based catalysts, (**b**) XPS test results of spent SCR catalysts after sintering into ceramics, (**c**) catalytic performance test results of NiO-based catalysts. Reproduced with permission [70]. Copyright 2020, Elsevier.

**Figure 6 materials-15-07984-f006:**
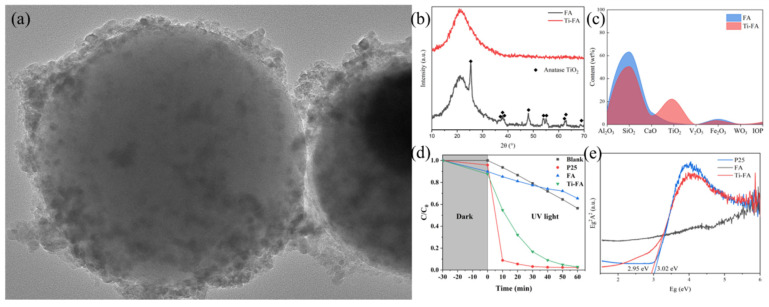
(**a**) Microscopic view of Ti-FA surface by TEM, (**b**) XRD pattern, (**c**) XRF test results and (**d**) UV degradation of rhodamine B catalytic performance test results; (**e**) UV-DRS test results. Reproduced with permission [74]. Copyright 2022, Elsevier.

**Figure 7 materials-15-07984-f007:**
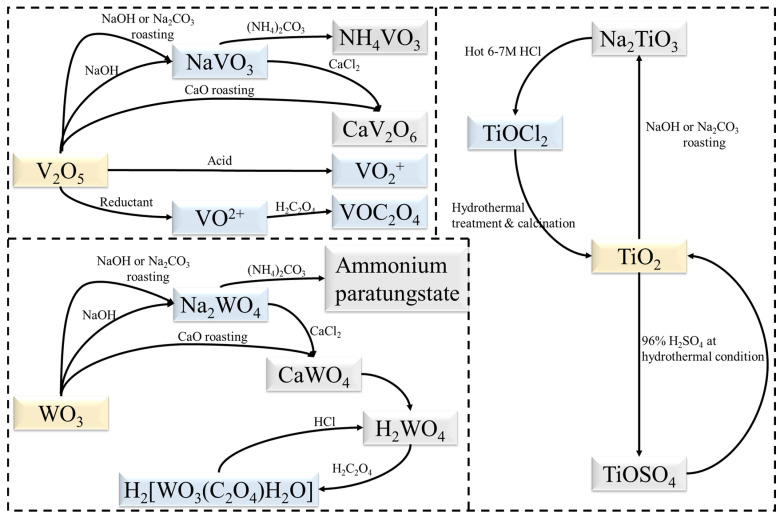
Transformation relationships of substances during extraction.

**Table 1 materials-15-07984-t001:** Regeneration process for SCR catalysts.

Process	Testing Condition	Performance	Ref.
Mixed acid immersing, NaOH immersing, CH_3_COOH immersing	0.2% NO_2_ + 0.2% NH_3_, 2 L min^−1,^ 400 °C	NO_X_ conversion = 73.0% (used: 17.1%, virgin: 89.9%)	[36]
H_2_SO_4_ immersing	0.05% NH_3_ + 0.05% NO, 0.2 L min^−1^, 400 °C	NO_X_ conversion ≈ 100% (comparable to fresh)	[37]
Ultrasonic water immersing, ultrasonic acid immersing, NH_4_VO_3_ impregnation, roasting, and loading	335 mg m^−3^ NO + 513 mg m^−3^ NO_2_, 1L min^−1^, 230 °C	NO_X_ conversion = 93.5% (higher than fresh)	[34]
Ultrasonic water immersing, ultrasonic EDTA+LAS immersing, and roasting	0.05% NH_3_+0.05% NO, 0.1 L min^−1^, 400 °C	NO_X_ conversion = 86% (poisoned: about 40%, fresh: nearly 100%)	[38]
Ultrasonic TMT103 immersing, ultrasonic EDTA+LAS+H_2_SO_4_ immersing, Ce(NO_3_)_3_·6H_2_O impregnation, and roasting	0.08% NH_3_+0.08% NO, 0.1 L min^−1^, 400 °C	NO_X_ conversion = nearly 99% (comparable to fresh)	[35]

EDTA: ethylenediaminetetraacetic acid. LAS: dodecylbenzene sulfonic acid. TMT103: 2,4,6-trimercaptotriazine.

**Table 2 materials-15-07984-t002:** Extraction process for spent SCR catalysts.

Process	Form of Ti	Leaching Efficiency of V	Leaching Efficiency of W	Ref.
(NH_4_)_2_CO_3_+H_2_O_2_ leaching	Anatase TiO_2_	98%	99%	[45]
NaOH leaching	Anatase TiO_2_	40.0~51.0%/66.8~69.8%	44.6~57.3%/38.0%~52.6%	[50]
NaOH leaching	In residue	95.76% on average	98.63% on average	[51]
NaOH leaching	Anatase TiO_2_	82.4%	54.3%	[52]
HCl leaching	Anatase TiO_2_	52.5%	0.1%	[52]
HNO_3_ leaching	Anatase TiO_2_	42.2%	0.0%	[52]
H_2_SO_4_ leaching	Anatase TiO_2_	69.0%	0.6%	[52]
NaOH leaching, H_2_SO_4_ leaching, carbothermic reduction, flotation operation, and electrochemical process	Metallic Ti (purity of 99.5 wt%)	91%	67%	[6]
NaOH+NaCl roasting, and water leaching	In residue	93.25%	99.17%	[53]
K_2_CO_3_ roasting, water leaching, Si removal, and CaCl_2_ precipitation	Potassium titanates	99.65%	96.89%	[54]
Na_2_CO_3_+NaCl roasting, and water leaching Resin enrichment, NaOH stripping, NH_4_Cl precipitation	Nano TiO_2_ and sodium titanate nanorods	94.9% of leaching 93.4% of precipitation	95.5% of leaching 96.2% of crystallization	[49]
Na_2_CO_3_ roasting, water leaching, and HCl leaching	TiOCl_2_	99.3%	99.2%	[55]
Na_2_CO_3_ roasting, and water leaching	Anatase/rutile TiO_2_	>99%	>99%	[48]
Roasting, H_2_C_2_O_4_ leaching, repeat, HCl precipitation, and roasting	Anatase/rutile TiO_2_	None	87%	[56]
Soda roasting, and water leaching Aliquat 336 extraction, and NaOH+NaCl stripping	In residue	>99.9% of extraction 71.31% of stripping	>99.9% of extraction 71.31% of stripping	[24]
NaOH+Na_2_CO_3_ roasting, HCl immersing, and calcination	Rutile/anatase TiO_2_ nanosphere	None	None	[7]

Aliquat 336: trioctylmethylammonium chloride.

## Data Availability

The data used to support the findings of this study are included in the article.

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
