# Peer review of "A Review on Resource Utilization of Spent V-W-Ti Based Selective Catalytic Reduction Catalysts"

_materials, 2022, doi:10.3390/ma15227984_

Round 1
Reviewer 1 Report
Developing technology for the regeneration of spent SCR catalysts and the processes used for elemental extraction is highly important for mitigating toxic elements (e.g., V, As, Pb, etc.) from entering soil and water bodies to avoid secondary pollution, and the key is to develop an efficient approach with economic benefits. In this review, the authors discussed the progress in research on spent SCR catalysts, regeneration, extraction of elements from spent SCR catalysts, conversion into photocatalysts and finally suggested a new resource of photocatalyst conversion processes with higher atomic efficiency. The review article is concise and well written with adequate background discussions and possible approaches. I would recommend this review for publication in the Materials.
Author Response
It is a great pleasure to receive your positive comments and thank you for your recognition of our work.
Reviewer 2 Report
This works deals with the review of the recovery of spent SCR catalyst. The reading is very interesting, it is well structured and written, and covers a reasonable period. In my opinion, it deserves to be published in Materials in its actual format since just small mistakes have been found such as:
Subscripts must be correctly included in the formulas in the introduction and references sections.
The edition must be checked, and some mistakes have been found (see Figure 2Error! Reference source not found. for SEM images).
Page 8 line 274, sodium titanate instead of sodium titanite, the same in Figure 4 footnote
Page 11 line 327 SiO44- instead of SiO42-
Author Response
It' s a great pleasure to receive your recognition and thank you for your careful work in pointing out the errors in our manuscript, which have now been corrected. Upon your recommendation we have also removed the "Error! Reference source not found" that appeared in line 224 (Page 7).
Reviewer 3 Report
In my opinion, Manuscript “materials-1948539” needs minor revision considering following issues:
First, authors should explain the reason for and advantage of the current review over previously published review (see ref*).
[*]-Zhang, Qijun, Yufeng Wu, and Haoran Yuan. "Recycling strategies of spent V2O5-WO3/TiO2 catalyst: A review." Resources, Conservation and Recycling 161 (2020): 104983.
2- Crustal abundance of titanium is high, and it is the 9th most abundant element on earth. Vanadium is also the 22nd most abundant element in the earth's crust (the fifth most abundant transition metal). The only rare metal in the original V-W-TiO2 SCR catalyst is tungsten. Therefore, I recommend modifying the introduction (first paragraph, term (ii), page 2-line 56). SCR catalyst supports are also made of ceramic or silica (materials abundant on earth). SCR catalysts can also contain other elements, but they come from combustion exhaust and their extraction is not reviewed in this article.
The manuscript should be checked to remove the language, typos, and formatting errors in the text. Example of problems:
a-page6- line 100: Figure 2Error! Reference source not found.
b-page7-line224
Reviewer 4 Report
Everything is fine, it is an interesting and informative review on the actual topic, it is worthy publishing.
I would just recommend very minor revision, regarding the abbreviations.
For example, Table 1 contain abbreviation "EDTA+LAS" which is not disclosed anywhere else in the text. The same for "TOPO, TIBPS, Aliquat 336 and HDEHP" (page 6, line 212).
Authors should also pay attention to some errors during production of pdf file of manuscript. These errors relate to incorrect processing of references to figures in the text (see messages like "Figure 2Error! Reference source not found" at page 6 line 200 and page 7 line 224). This is a responsibility of authors to check the correct conversion of initial text into pdf.
Author Response
Thank you for your recognition of our work and for your valuable suggestions. We have now checked all the abbreviations in the text and modified them accordingly (see Table 1 and Table 2 footnotes as well as page 6 line 220). Moreover, the text of 'Error! Reference source not found' has been removed.